PERSPECTIVE

# Functional dynamics in framework materials

Simon Krause [1,3✉] & Jovana V. Milić[2,3✉]

Dynamic crystalline materials have emerged as a unique category of condensed phase matter that combines crystalline lattice with components that display dynamic behavior in the solid state. This has involved a range of materials incorporating dynamic functional units in the form of stimuli-responsive molecular switches and machines, among others. In particular, it has been possible by relying on framework materials, such as porous molecular frameworks and other hybrid organic-inorganic systems that demonstrated potential for serving as scaffolds for dynamic molecular functions. As functional dynamics increase the level of complexity, the associated phenomena are often overlooked and need to be explored. In this perspective, we discuss a selection of recent developments of dynamic solid-state materials across material classes, outlining opportunities and fundamental and methodological challenges for their advancement toward innovative functionality and applications.

In crystalline materials, the function is conventionally perceived from structural, electronic, and compositional points of view. However, solid-state materials feature functionality beyond their static compositional arrangements through various mobile distortions that can lead to emerging functions. Such intrinsic dynamics in crystalline materials involve lattice vibrations, phonons, and other coupled structural degrees of freedom that can produce collective functional responses. This has been increasingly relevant across disciplines to design and control crystalline materials with specific stimuli-responsive dynamics and controlled molecular motion. Such efforts have led to a wide range of dynamic crystalline materials (DCMs) known by different terms. Amphidynamic materials (ADMs)[1,2] is a term coined by M. Garcia-Garibay to describe condensed phases that combine crystalline order and liquid-like dynamics, built with lattice-forming elements linked to components that can undergo motion[3,4]. ADMs have emerged as a category of condensed phase matter incorporating dynamic functional units, including molecular switches and more complex molecular machinery, such as motors and shuttles[5]. As a certain degree of dynamic molecular functionality is preserved in the solid state, more generally, DCMs exhibit structural dynamics beyond conventional structural rearrangements in condensed matter, i.e., phase transitions or lattice deformations. Instead, they feature robust dynamics, a term coined by Stoddart and Yaghi to describe intrinsic dynamics within a static or rigid framework[6]. In contrast, Kitagawa defines such materials as soft porous crystals that can feature both deformable lattices and local, responsive dynamics of DCMs[7]. The research on DCMs develops at the intersection of several disciplines, including organic synthesis, solid-state chemistry, crystal engineering, reticular chemistry, supramolecular chemistry, and molecular mechanochemistry, complemented with a vast array of synthetic, analytical, and computational tools to describe these systems both experimentally and theoretically which we do not review here in detail[3,8–17].

In this perspective, we discuss DCMs from the standpoint of criteria that define them and opportunities presented to realize functional dynamics across classes of different (molecular) framework materials[18]. In particular, we outline unique characteristics describing functional dynamics based on representative examples and consider methodological and other challenges for analyzing dynamic functions across length scales. Given the various existing terminologies,

[1] Max Planck Institute for Solid-State Research, Stuttgart, Germany. [2] Adolphe Merkle Institute, University of Fribourg, Fribourg, Switzerland. [3] These authors contributed equally: Simon Krause, Jovana V Milić. ✉email: s.krause@fkf.mpg.de; jovana.milic@unifr.ch

we describe these materials with responsive dynamic features and functionalities beyond lattice vibration and deformation as DCMs. We further address the importance of interdisciplinarity in providing a fundamental understanding of the pressing challenges, hoping to inspire researchers and new perspectives for this fascinating field.

## Phenomenology of functional dynamics

The fundamental characteristics governing DCMs across material classes and related functional dynamics require several considerations associated with the role of motion. This applies to different crystalline (molecular) framework materials defined by periodic assemblies of various nodes and/or (molecular) linkers that are relevant in this context (Fig. 1a)[18], such as metal-organic frameworks (MOFs), covalent organic frameworks (COFs), and coordination polymers (CPs), but also hydrogen-bonded organic frameworks (HOFs), zeolites, periodic mesoporous organosilicas (PMOs), and hybrid perovskites, among others, that have been underrepresented as prospective DCMs. These organic (e.g., COFs, HOFs) and hybrid organic-inorganic (e.g., MOFs, CPs, and hybrid perovskites) framework materials have attracted attention due to the capacity of their network structures to incorporate dynamic molecular components either within void space or into the framework backbone (Fig. 1b). To this end, it is essential to differentiate degrees of freedom (i.e., rotational, translational, and conformational, etc.) in such molecular frameworks and distinguish between random, restricted, controlled, and directional motion (Fig. 1a). Finally, the control of DCMs necessitates considering bottom-up assembly strategies and specific stimuli to activate and drive functional dynamic responses, along with their complex and cooperative operation.

In general, molecular systems exhibit intrinsic motion defined by the molecular degrees of freedom and their mobility is dictated by temperature. When embedded into a crystalline lattice, some degrees of freedom are constrained, thus drastically changing their dynamic nature and limiting intramolecular motion. However, if certain molecular constituents are not directly involved in the lattice formation, they may exhibit **random motion** comparable to the gas phase, although immobilized within a crystal lattice. As a result, crystalline networks can have domains of high order with low entropy (i.e., the backbone of the framework) as well as low order and high entropy (i.e., flexible side groups or counterions) interconnected within a multidimensional framework. For instance, Pallach et al. have recently demonstrated that flexible side chains in MOFs can act as dispersion energy donors that show solvent-like behavior, which enables drastic changes in the corresponding energy landscape of a MOF-5-type framework. As a function of temperature, the dynamic dispersive interactions can deform the static backbone of the framework in response to the intrinsic dynamics of the side chains[17]. Such functional dynamics based on random molecular motion is relevant for other hybrid organic-inorganic framework materials, such as hybrid metal-halide perovskites[18–22]. They have shown a unique capability to incorporate various organic species within their crystalline lattice that is otherwise primarily defined by the inorganic metal-halide framework[18–23]. As a result, the overall structural properties and functional dynamics could be controlled by an interplay of interactions between the organic and inorganic components in response to various external stimuli[23]. For instance, this enables reversible pressure-induced mechanochromism in these materials and, since recently, the integration of stimuli-responsive molecular components within the perovskite scaffold, opening a path towards multifunctional materials[17,18,22]. Similarly, cations in aluminum-rich zeolites exhibit dynamics within the cavities, blocking the transport of guest species.[24] In addition, light-controlled zeolite membranes have shown dynamic solid-state functionality. For instance, azobenzene incorporation into the zeolitic cavities was used to tailor its permeation[25], illustrating the capacity of other frameworks in realizing functional dynamics within pore cavities.

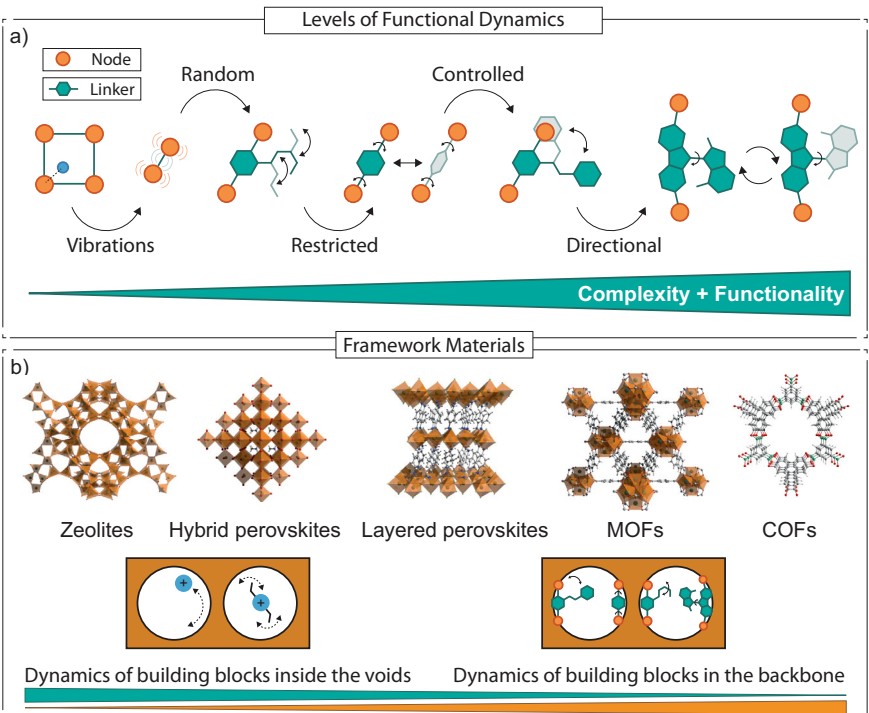

**Fig. 1 Levels of functional dynamics and framework materials.** Structural representation of functional dynamics expressed through various degrees of freedom and the associated random, restricted (i.e., constrained, and (controlled) directional motion (**a**) in representative (molecular) framework materials and underlying dynamics of building blocks either inside the void space or within the framework backbone (**b**).

Such complex materials and their intramolecular motion can be tailored to achieve specific functionalities by **restricting certain degrees of freedom** through molecular design and incorporation into the framework backbone. Among the most popular examples are molecular rotors predominantly based on para-substituted cyclic molecules (e.g., aromatics) with low torsional barriers featuring restricted non-directional motion[13]. While intermolecular (e.g., Van der Walls, π-π) interactions drastically enhance the barrier for rotation in densely packed molecular crystals, spatial separation of adjacent rotating molecules in molecular frameworks allows unrestricted rotational dynamics within a rigid 3D lattice[1–3,26]. For example, Perego et al. observed enhanced rotational frequency of bicyclo[1.1.1] pentane-1,3-dicarboxylic acid even at temperatures of 2 K in a Zr-based MOF due to its threefold rotational symmetry that exhibits a geometric mismatch concerning the fourfold-symmetry of the framework-forming node[27]. Rotational motion is also observed in PMOs depending on the packing density of the rotors[28]. Similarly, intramolecular dynamics permits restricting molecular motion by mechanically interlocking two or more molecular species without any covalent intermolecular interaction. This leads to the formation of catenanes[29] or rotaxanes[30] that are known to exhibit rotational (i.e., the ring rotates around a ring or strut) or translational (i.e., ring slides along a linear or circular track) motion even when embedded in the framework[18,31]. While this illustrates the effect of facilitating specific dynamics in the solid state, the underlying motion is dictated by the molecular structure of the dynamic unit and its surroundings, driven by temperature and only to some degree directional in its nature.

To achieve **controlled non-directional motion**, artificial molecular switches can be externally controlled through rational molecular design[32]. This can, for instance, be achieved by incorporating photoswitches into the framework backbone that change their structure in response to light. Incorporating azobenzenes is a typical example of allowing for dynamic control in MOFs[33]. However, the photoswitching mechanism strongly depends on the framework structure[34] and the switching of azobenzene moieties might exhibit a different mechanism when the switch is embedded within the framework through non-covalent complexation as compared to being attached to the framework backbone[35]. Williams et al. showed that the photoisomerization of diarylethenes and spiropyranes could also be manipulated by the framework confinement[36]. In addition to photoinduced dynamics, the translational motion could be achieved electrochemically in bistable redox-active catenanes attached to the internal surface of a Zr-based MOF[37].

While achieving stimuli-induced (non)directional motion is desirable in functional materials, **continuous (oscillatory) (uni) directional motion** is a prerequisite for a system to perform work comparable to macroscopic machines[38]. Unlike the previous examples, these dynamic systems can transform specific energy inputs into unidirectional motion through a ratcheting mechanism[38]. Feng et al. showed that MOF-anchored rotaxanes[39] could facilitate the unidirectional transport of rings onto a polymer strand via redox cycling[40]. Although these rotaxanes were attached to 2D MOF sheets rather than embedded into a 3D crystalline solid, and their translational motion is not unidirectional once threaded on the strut, the fundamental concept illustrates the feasibility of stimuli-responsive unidirectional motion in this context that can be extended to other material classes. Exemplary of such functionality are light-driven molecular motors, especially overcrowded alkene-based molecular motors. Light-induced isomerization followed by a thermal relaxation step allows for a ratcheting mechanism that results in unidirectional 360° rotation[41]. For instance, Danowski et al. embedded pyridine-functionalized motors into the backbone of

MOFs while preserving unidirectional rotation[42,43]. This provides the only example of unidirectional continuous responsive motion in crystalline solids. However, recent work by Gao et al. and Kathan et al. demonstrated how unidirectional rotation can result in the winding of polymer strands, resulting in macroscopic contraction of the molecular or polymer species[44,45]. The underlying effects of such interlinked dynamics on a crystal lattice in DCMs and their functionality remain to be explored.

## Establishing dynamic materials and functional dynamics

Achieving functional dynamics in a regular lattice necessitates controlling the bottom-up assembly and the stimuli-responsiveness of the framework materials. The **bottom-up (self-)assembly** of intrinsically dynamic building blocks in (molecular) framework materials is a promising strategy toward this goal. It can involve crystallizing dynamic molecular building blocks into molecular crystals while maintaining intermolecular dynamics[1,2]. Although many such dynamic molecular crystals are already known, their overall arrangement often lacks stability, and synthetically controlling their structure, porosity, and growth into extended functional materials remains challenging[9]. Materials such as MOFs, COFs, and hybrid perovskites have been considered as platforms in which dynamic building blocks can be co-assembled with secondary organic or inorganic components via coordination or covalent bonds to form robust 2D or 3D lattices. The framework structure can be predefined by the geometry of the building blocks, providing a vast structural space that can, to some degree, be predicted[46,47]. These frameworks often feature intrinsic porosity that facilitates intermolecular separation of building blocks for unrestricted dynamics and can also support host-guest interactions. However, porosity does not guarantee that unrestricted dynamics can occur. For example, Stähler et al. have incorporated light-driven molecular motors in the backbone of COFs which exhibit pore sizes that exceed the diameter of the motor threefold[48]. Nonetheless, the stacking of 2D COF sheets in the third dimension could still prevent free molecular motion through interactions between the rigid building blocks of the backbone and adjacent dynamic moieties. These challenges stimulate ongoing investigations on controlling the assemblies of DCMs without compromising their functional dynamics.

While precise structural arrangements permit controlling molecular dynamics in DCMs, control over such motion is possible through **external stimulation** that enables functional response beyond temperature-controlled random motion and towards (uni)directional motion. To this end, different stimuli have been applied[35–37,49–51], including chemical processes (e.g., pH changes or chemical fuels),[37,51] electrical charge[37], light[35,43], and mechanical pressure[19,52]. The resulting DCM response has predominantly focused on molecular building block dynamics, overcrowded alkene-based molecular motors in particular, without considering the impact on the extended material properties, especially in bulk materials. For example, in larger crystals or thicker samples of bulk films or powders, the effect of stimulation by electric or light bias strongly depends on the penetration depth. A recent analysis of the photoswitching in MOFs illustrated that the penetration depth is limited to only a few 100 μm or less, restricting the bulk material functionality[35]. However, incorporating a dynamic molecular species in an ordered lattice allows alternative ways to stimulate the resulting dynamics. For instance, Moggach and co-workers show that the rotational dynamics of linkers can be manipulated by hydrostatic pressure applied to a single crystal[49,50]. Similarly, two-dimensional (layered) hybrid perovskites incorporating various organic cations as spacers templating inorganic perovskite slabs feature reversible and anisotropic pressure-induced structural changes, as well as mechanochromism in moderate pressure up to

0.35 GPa[19,52]. The anisotropic nature of crystals also allows for specific stimulation via magnetic or electric fields[53]. However, it is still debated whether currently applied field strengths can drive fast dynamics of polar linkers in DCMs[54]. On the other hand, most framework materials' porosity allows molecular building blocks to interact with guest species, thereby enabling guest-responsiveness or chemical activation, such as via changes in pH[51]. Electrochemical switching was also demonstrated, which requires the transport of electric charge and ions to the dynamic building block, which can be facilitated by intrinsic porosity[37]. Similarly, Danowski et al. showed that energy transfer could stimulate framework-embedded light-driven motors via a secondary porphyrin linker excitation[43]. This interplay of two different organic linkers relies on the close spatial proximity of both species. It provides an alternative mechanism for perpetuating the stimuli-induced response in framework materials.

Such interplay of molecular building blocks in DCMs allows for different levels of **cooperativity**, particularly in amplifying dynamic responses toward macroscopic effects. Some of the earliest and most representative examples of solid-state molecular machinery with correlated motion are molecular gears transmitting motion from one element to another, including rotary arrays, molecular pumps, and motors[3]. Inspired by an imaginary "molecular clockwork", Wilson et al. explained how a rotaxane and a rotor could interact when co-crystallized in a Zr-based 3D MOF to illustrate this approach[55]. Similarly, Gonzalez-Nelson et al. demonstrated coupled rotational dynamics of adjacent linkers in an Al-based MOF as a function of their functionalization[56]. Moreover, Perego et al. made a similar observation by investigating cascade dynamics in a MOF comprised of two different rotor linkers, which exhibit distinct rotational behavior after a phase transition of the flexible lattice and upon adsorption of carbon dioxide[57]. Given that MOFs and other related framework materials can exhibit structural transformations of the framework (e.g., folding, expansion, or shrinkage of the lattice, etc.), the cooperative effects of the linkers provide the opportunity to control functional dynamics on the framework level that is relevant for macroscopic effects.

This further emphasizes the **levels of complexity** that need to be considered in the design, analysis, and application of dynamic framework materials, including their structure and composition (Fig. 2a). Specifically, with respect to structural complexity, a rich pool of potential building blocks (i.e., nodes, counterions, and linkers; Fig. 1) results in a seemingly endless number of framework topologies; yet, selecting appropriate topologies enables bringing different dynamic molecules in close proximity to intermolecular cooperativity[55]. However, a detailed understanding of the structural arrangement is required for this approach to be effective. Kolodzeiski et al. showed in a computational study that collective motion poses specific geometric requirements for the assembly, in particular, if the underlying building block is of lower symmetry[58]. Furthermore, multivariant frameworks or solid solutions enhance the compositional complexity by incorporating multiple chemically different but geometrically similar building blocks in a single crystal[59]. For example, Inukai et al. showed that the rotational behavior of bipyridine rotors could be tuned by altering the composition of a solid solution[60]. In contrast to structural topologies with distinctly defined positions of building blocks, the compositional control in solid solutions is, so far, less precise concerning local order but may provide a powerful design tool if sequencing would become more controllable[61]. Yet, it allows manipulating dynamics beyond the unit cell of interest to future applications[28]. This, in turn, requires the assessment and description of dynamics across length and time scales. Given the limited number of studies that attempt to describe multifaceted dynamics in complex framework materials, it becomes evident that an accurate description and illustration of the increasing number of relevant parameters and the resulting multidimensional landscapes (Fig. 2b) is challenging[57].

## Functional dynamics across length and time scales
One of the significant challenges to realizing the potential of functional dynamics in molecular frameworks is monitoring the underlying phenomena across length and time scales (Fig. 2).

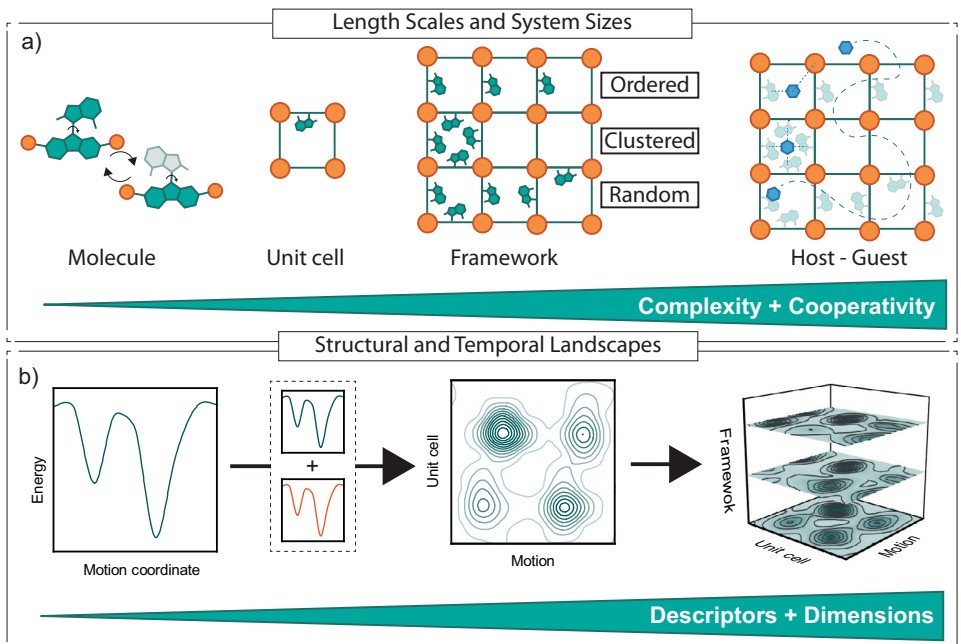

**Fig. 2 Complexity of functional dynamics in framework materials.** Schematic of the multidimensionality of the energy landscapes that can describe functional dynamics across **a** different length scales and systems, from the molecular level through the unit cell to the framework level, including host-guest systems, which **b** need to be complemented by temporal descriptors.

This requires a combination of theoretical and experimental techniques that involve structural assessment as well as the analysis of characteristics of the materials via a range of spectroscopic and spectroelectrochemical techniques (Fig. 3)[61–65].

In particular, structural characteristics of DCMs are commonly analyzed using X-ray diffraction techniques in conjunction with theoretical models, as well as pair distribution function analysis[35,48]. Furthermore, the analysis of anisotropic displacement parameters available from variable temperature single crystal X-ray diffraction is relevant, along with the use of frequency and temperature-dependent dielectric spectroscopy and inelastic neutron scattering for ultrafast dynamics[64,65]. Modern time-resoled electron diffraction experiments allow following light-induced processes on the nanoscale, drastically enhancing the time and space resolution of crystallographic methods[66]. Although these methods permit the assessment of crystallographic characteristics, establishing accurate modes in terms of extended structural complexity and dynamics levels is an ongoing challenge. This could be addressed by considering complementary techniques, such as Raman/IR and solid-state NMR spectroscopy, which remain underrepresented in this context despite the capacity to offer atomic-level insights[20–22]. This mainly refers to NMR crystallography, which is of increasing interest yet also limited by the lack of reliable theoretical models to analyze the underlying dynamics[22,67].

Another limitation refers to the challenge of reliably introducing and applying external stimuli, such as electric or light bias, during measurements without interfering with the data acquisition. The limited availability of in situ structural characterization techniques of these functional materials during operation (i.e., irradiation or voltage bias) remains a pressing challenge for further advancement. Furthermore, other spectroscopic methods that assess optoelectronic properties and indirectly provide input on functional dynamics are relevant. This involves steady-state and time-resolved UV-vis absorption and photoluminescence spectroscopy and spectroelectrochemistry, as well as transient absorption and THz spectroscopy for complementary inputs[68]. Although the currently available toolbox of methodologies (Fig. 3) allows capturing fast dynamic processes from the molecular to the macroscopic level, direct access to the structural information and inter- and intramolecular processes during exposure to external stimuli remains challenging. In most experimental setups, the motion of the dynamic species can be slowed down by variation in temperature, allowing to extrapolate activation barriers at higher temperatures which would otherwise be out of the sensitivity range of the respective method. This stimulates further efforts to complement the experimental findings with theoretical insights and modeling.

However, similarly to the structural analysis, one of the challenges for interpreting the experimental data is the lack of reliable theoretical models[38] and computational methods[61] especially those that consider dynamics over longer time scales and complex assemblies beyond an individual dynamic molecule. In particular, this refers to the translation of basic thermodynamic and kinetic models to describe the complex landscapes and processes in molecules[69] toward more complex materials on larger length scales, which might take inspiration from engineering disciplines but need to consider the underlying functional principles of dynamics at the molecular scale[70]. The capacity to monitor the dynamics in DCMs opens the path to exploiting other phenomena that can result in functionalities dictated by dynamics. This, for instance, involves dynamic host-guest properties. By comparing the rotational frequency of a rotor in a guest-free and guest-filled MOF, Jiang et al. could estimate the viscosity of the solvent in the pore[71]. Continuous threading of rings on a rotaxane grafted onto the surface of a Zr-MOF resulted in redox-responsive adsorption based on mechanical bonds, termed mechanisorption[40]. Recent works on photoswitching illustrate the possibility of manipulating permeability for membrane separation[72] or guest uptake[40]. Gu et al. describe activated transport by pore window agitation via a molecular flip within a MOF that displays unique adsorption properties as a function of temperature[73,74]. Demonstrated in a computational study that incorporating dipolar gates might even allow to selectively open and close pore channels dynamically by using an electric field Tam et al.[74]. However, these phenomena are based on a molecular switching behavior without considering continuous (unidirectional) dynamics in DCMs that have only been the subject of computational investigations. In contrast, Kolodzeiski et al. investigated cooperative shuttling in a rotaxane-based MOF and the collective behavior that can arise from such a complex system[75]. However, among the most intriguing and thought-after properties is directed activated diffusion (i.e., pumping) via the rotational motion of molecular motors in porous frameworks. Evans et al. outlined the physical feasibility of such a process in a computational model and illustrated the decisive factors to realize it[76]. While these attempts have been made by demonstrating activated transport via the rotation of motors embedded in ion channels[77] or MOFs[78] directional molecular transport of guest species in porous hosts, e.g., via unidirectional rotation as predicted in simulations[76] is yet to be discovered in real world.

## Summary and outlook

Research on (amphi)DCMs has been gradually emerging over the last few decades, and its advancement requires an interdisciplinary approach considering both molecular and solid-state properties across material classes (Fig. 4). The representative examples of DCMs in various molecular frameworks suggest that sophisticated molecular design and synthetic strategies to access these materials have already been established, yet several challenges remain in terms of both their fundamental understanding and future application.

One of the limiting factors refers to the toolbox for analyzing dynamics in the solid state on the bulk, thin film, or single crystal level, especially from the perspective of in situ structural characterization methods in response to external stimuli such as

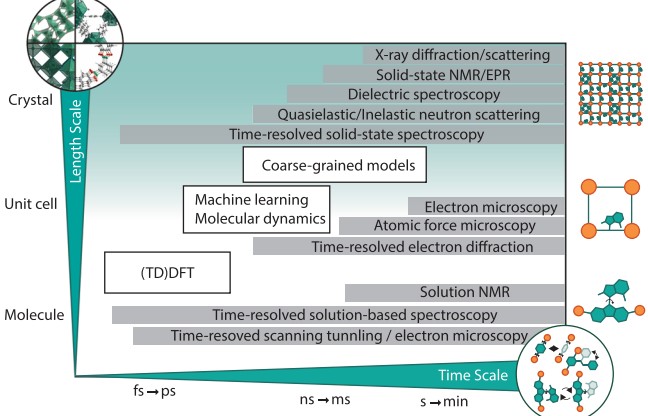

**Fig. 3 Investigation of functional dynamics of framework materials in space and time.** Representative experimental (gray bars) and computational (white boxes) tools to study time-dependent processes in molecules and DCMs as a function of spatial and temporal resolution. This figure was inspired by ref. [63–65] and does not provide a complete list of methods to study dynamics in framework materials.

**Fig. 4 Interdisciplinary efforts to address functional dynamics in framework materials.** Illustration of dynamics in framework materials and the associated challenges and opportunities that require cooperative and interdisciplinary approaches across disciplines and material classes. Adapted with permission from ref. [82] under CC-BY 4.0 licence.

light[78,79]. Moreover, theoretical tools to analyze, describe, and potentially predict dynamic properties across length and time scales are still limited[61]. While functional dynamics opens the way to unique phenomena, reaching beyond the conventional boundaries of research disciplines and material classes associated with solid-state dynamics is essential to overcome some pressing challenges[80]. Dynamic features present an intrinsic characteristic of any solid-state material and should be considered with equal importance compared to the underlying structure or composition of the material[81]. The sooner we consider dynamics holistically as functional aspects of a material, the better we will understand and exploit the underlying phenomena. We hope this emerging research area stimulates more collaborations across disciplines[82] to address functional solid-state dynamics and realize its innovative potential.

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

## Acknowledgements

S.K. acknowledges funding by the German Research Foundation (Deutsche Forschungsgemeinschaft, DFG) within the collaborative research center CRC 1333 (Project No. 358283783), the Carl-Zeiss Stiftung (Nexus program) and the Fonds der Chemischen Industrie. J.V.M. acknowledges Swiss National Science Foundation (SNSF) PRIMA fund no. 193174 for financial support. The authors are grateful to the "Dynamic Materials, Crystals, and Phenomena" community contributing to the DynaMiC conference, whose contributions to this fascinating field and discussions throughout the years inspired this perspective.

## Author contributions

S.K. and J.V.M. conceptualized the perspective, reviewed the literature, and contributed equally to writing and revising the manuscript.

## Competing interests

The authors declare no competing interests.
