## [Peer Review File · Communications Chemistry]

Reviewers' comments:

Reviewer #1 (Remarks to the Author):

The present contribution is an overview of dynamic materials which exploit low density or porosity for achieving internal mobility in solids. I believe this subject, including molecular rotors and motors, is on the front of the present day research.

To improve the manuscript, I am listing a few comments and observations:

1) The term amphidynamic recalls a restricted research area and is not fully aligned with the examples discussed in the text. It was used at the beginning of the Abstract. It should be substituted by Dynamics in crystalline solids or similar expressions indicating molecular dynamics in Materials.

The paradigm of ADM is the motif of the Introduction and applied to a few families of porous and non-porous materials, which were not labelled in this way by the authors of the original articles. This seems an attempt to constrict artificially the recognition of dynamics in crystalline solids into a single label. Much wider breath would be ensured by a less specific context to the benefit of clarity. Given the complexity reached by these category of dynamic materials, the term amphidynamic is misleading nowadays: dynamics is more than enough to define mobility in solids. Contrarily, the concepts of molecular and crystallinity, are missing in the ADM acronym, which, together with amphidynamics, should be thus removed from the entire manuscript.

2) Continuing on the definition of the field, the title: Functional Dynamics in Framework Materials is not attractive, because the expression Framework materials is not well defined in literature and sounds rather exotic. Additionally, perovskites, extensively treated by the authors, are not generally considered as Frameworks.

3) The concepts of unidirectional and directional should be better distinguished and discussed, because they are recurrent in the manuscript: the theme is hot and deserves to be inscribed within neat boundaries to avoid confusing sentences. Experimental and theoretical results on this issue must be always distinguishable.

Reviewer #2 (Remarks to the Author):

This is an interesting perspective covering the field of functional dynamics in Framework materials. The subject coverage seems to be good and based on interesting from a wide range of authors. One concern is that the presentation is likely to miss the intended audience, which I assume should be relatively broad, covering chemists, physicist and materials scientists not in this field, as well as advanced undergraduate and graduate students. I will enumerate my comments and suggestions as follows:

(1) Most figures are not as useful as they could be. The figure in the abstract does not convey a useful message. None of the figures illustrates what ADM framework materials might look like. Figure 1a is probably the most useful in the entire perspective, but it could be much better if it showed how those dynamic groups connect to the extended frameworks in Figure 1b.

(2) How does an ADM zeolite look like? Are there any? It seems that what the authors may have in mind zeolites that undergo phase changes upon the absorption of some guests, like ZSM-5 and para-xylene?

(3) The use of the term “omnidirectional” seems inappropriate. Omnidirectionality implies that a given molecular entity can explore all the degrees of freedom that it has in the gas phase, or in a non-interacting liquid. The fact is that this is never the case when constrained environments reach the nanometer scale. Furthermore, the example shown with a floppy side chain linked to the framework cannot be omnidirectional as shown by the fact that it they are attached by a fixed permanently directional bond and cannot experience all rotational degrees of freedom. There must be a better term to describe this case. Constrained dynamics? Disordered dynamics?...

(4) Figure 2 is not useful if it is not more explicit as to what it all means, with the trajectories and dynamics of the corresponding elements. It is not clear what it means for a dynamic framework to be multidimensional. The message in Figure 3 is also poorly represented.

(5) In page 3 second paragraph the authors refer to “strong” intermolecular pi-pi interactions as resulting in large barriers, which are not present in framework materials. In fact, pi-pi are not considered strong. Subsequently the authors offer a single example to make the comparison between molecular crystals and frameworks and they select a non-aromatic structure, bicyclo[2.2.2]octane. However, browsing through the literature one can find examples of molecular crystals with aromatic phenylenes rotators experiencing barriers lower than those of methyl groups in the gas phase, and one can also find open frameworks with rotators that have to surmount a very large intrinsic barriers, such as the iconic MOF 5. A recent JOC review article explicitly covers variations between different types of solids based on intrinsic barriers, free volume, and correlated dynamics that provides a general framework for analysis.

(6) In the last paragraph in Figure 4 the authors give a list of stimuli reported to influence the dynamics or properties of framework materials. It would be very useful for the interested readers to have references to those reports.

(7) The section on the assessment of functional materials across length scales in page 4 describes some of the analytical tools to measure dynamics in extended frameworks and other solids. It seems that this section is missing some interesting tools. This include the analysis of anisotropic displacement parameter (ADP) available from variable temperature single crystal x-ray diffraction, the information available from crystallographic disordered, which can be static or dynamic, and the use of frequency and temperature-dependent dielectric spectroscopy, and even inelastic neutron scattering for ultrafast dynamics. Giving the reader and idea of what method to use in what dynamic range would be greatly appreciated.

(8) The paragraph on dynamic cooperativity in dynamic crystals would not be complete without considering the 2020 Chemical Sciences essay on correlated motion and mechanical gearing by the Garcia-Garibay group.

To conclude, this reviewer feel that this perspective has great potential, but it needs a little work with the goal of becoming more scholarly, educational, and broad reaching.

Reviewer #3 (Remarks to the Author):

In the present perspectives article, some aspects of light-responsive framework materials are addressed. The focus is on one kind of molecular motors in COFs and MOFs. At first, it is not obvious why the focus was chosen, see below.

The perspectives-article addresses a dynamic field. However, I cannot recognize a valuable contribution to the field. Recently, many excellent review articles were published, like Ref. 14, 15, <https://doi.org/10.1002/anie.201900666>, <https://doi.org/10.1021/acs.chemrev.1c00528> and <https://doi.org/10.1016/j.xcrp.2021.100544>.

In comparison, the novelty, creativity and meaning of the present article are limited.

I reckon that the perspective may eventually published, but I have my reservations that a journal like Communications Chemistry is appropriate. A more specific journal is more appropriate.

Before resubmission, some issues need to be revised:

- The article is filled with strong catchphrases without providing scientific insights. At least some solid scientific insights need to be provided.
- The title does not match the content. A more accurate title is needed.
- The addressed dynamics focus on light-driven molecular motors of special kind. By checking the cited references, it is obvious the focus is on overcrowded-alkene-molecular-motors. This needs to be stressed in the text.
- Figure 1 shows zeolites and perovskites. To term them framework materials is uncommon.
- The dynamics in zeolites and perovskite are not discussed in the text or, at least, mentioned. Especially the early work of light-switched zeolite membranes seems important to provide a time line for the field of dynamic solid materials.
- The authors should explain what they mean with the term “mechanism”, used in Figure 3 and the abstract.
- The metaphors of the 4 climbers and the map in the abstract figure remain mysterious to me, even after spending some time trying to comprehend it. It should be revised.
- The several statements in the conclusion are hard to follow:
“One of the limiting factors refers to the toolbox for analyzing dynamics in the solid- state, especially from the perspective of in-situ structural characterization methods in response to external stimuli.” – The methods summarized in <https://doi.org/10.1021/acs.chemrev.1c00528> were used...
“While functional dynamics opens the way to unique phenomena, we believe that it is critically important to reach beyond conventional boundaries of research disciplines and material classes to overcome some of the pressing challenges.” – Which are?

Dear Reviewers,

Herein we submit a revised manuscript entitled “**A Perspective on Functional Dynamics in Molecular Frameworks**” in the special issue of *Communications Chemistry*. On behalf of the authors, we would like to thank the reviewers for their detailed analysis and critical remarks which helped us to improve the manuscript (COMMSCHEM-22-0553) substantially. The reviewers requested clarification of several points in the discussion and made critical remarks, which we have now addressed as well as substantially rewritten the manuscript. We submit two versions of the revised manuscript, one with highlighted changes and one without, while we provide the answers to the questions of the reviewers in detail within the following Response Letter.

Reviewer Comments

Reviewer #1

The present contribution is an overview of dynamic materials which exploit low density or porosity for achieving internal mobility in solids. I believe this subject, including molecular rotors and motors is on the front of the present day research. To improve the manuscript, I am listing a few comments and observations:

Response: We appreciate the reviewer's constructive remarks and support for the research subject in terms of its relevance at the forefront of today's research. We address their comments directly below.

1. *The term amphidynamic recalls a restricted research area and is not fully aligned with the examples discussed in the text. It was used at the beginning of the Abstract. It should be substituted by Dynamics in crystalline solids or similar expressions indicating molecular dynamics in Materials. The paradigm of ADM is the motif of the Introduction and applied to a few families of porous and non-porous materials, which were not labelled in this way by the authors of the original articles. This seems an attempt to constrict artificially the recognition of dynamics in crystalline solids into a single label. Much wider breath would be ensured by a less specific context to the benefit of clarity. Given the complexity reached by these category of dynamic materials, the term amphidynamic is misleading nowadays: dynamics is more than enough to define mobility in solids. Contrarily, the concepts of molecular and crystallinity, are missing in the ADM acronym, which, together with amphidynamics, should be thus removed from the entire manuscript.*

Response: We define the term "amphidynamic materials" following the original definition by Garcia-Garibay who coined it to describe "condensed phases that combine crystalline order and liquid-like dynamics, built with lattice-forming elements linked to components that can undergo fast motion" (*Proc. Nat. Acad. Sci. U.S.A.* **2005**, 102, 10771). This category of materials emerged to encompass solid-state systems that incorporate dynamic functional units beyond conventional structural rearrangements, such as phase transitions or lattice deformations, but other functional components that feature intrinsic dynamics in the form of molecular switches and machines referred to as "robust dynamics" (*Nat. Chem.* **2010**, 2, 439). This does not constitute an attempt to constrain the dynamics of crystalline solids in a single label and we specify this by referring primarily to the functional dynamics in molecular framework materials (*Acc. Chem. Res.* **2021**, 54, 1288). We clarify this in the revised manuscript. Given the various existing terminologies that can be unintentionally misleading, we describe these materials with dynamic features beyond lattice vibration and deformation as dynamic crystalline materials (DCMs), while welcoming other suggestions.

"Such efforts have led to a wide range of various dynamic crystalline materials (DCMs) known by different terms. *Amphidynamic materials (ADMs)*^{1,2} is a term coined by M. Garcia-Garibay to describe condensed phases that "combine crystalline order and liquid-like dynamics, built with lattice-forming elements linked to components that can undergo fast motion".^{3,4} ADMs have emerged as a category of condensed phase matter incorporating dynamic functional units, including molecular switches and more complex molecular machinery, such as motors and shuttles.⁵ As a certain degree of dynamic molecular functionality is preserved in the solid state, more generally, DCMs exhibit structural dynamics beyond conventional structural rearrangements in condensed matter, i.e. phase transitions or lattice deformations. Instead, they feature *robust dynamics*, a term coined by Fraser Stoddart and Omar Yaghi to describe intrinsic dynamics within a "static" framework.⁶" (Page 1)

"In this perspective, we discuss *dynamic crystalline materials (DCMs)* from the standpoint of criteria that define them and opportunities presented to realize *functional dynamics* across classes of different *molecular framework materials*.¹⁸ In particular, we outline unique characteristics describing functional dynamics based on representative examples and consider methodological and other challenges for analyzing dynamic functions across length scales. Given the various existing terminologies, we describe these materials with responsive dynamic features and functionalities beyond lattice vibration and deformation as DCMs. We further address the importance of interdisciplinarity in providing a fundamental understanding of the pressing challenges, hoping to inspire researchers and new perspectives for this fascinating field." (page 1)

-
2. *Continuing on the definition of the field, the title: Functional Dynamics in Framework Materials is not attractive, because the expression Framework materials is not well defined in literature and sounds rather exotic. Additionally, perovskites, extensively treated by the authors, are not generally considered as Frameworks.*

Response: We appreciate the reviewer's input regarding the title and we regret that they do not find it attractive. The terminology defining "framework materials" is well-defined and appropriately used in the manuscript to refer to crystalline materials defined by periodic assemblies of specific nodes and molecular linkers, which also includes hybrid organic-inorganic perovskite materials (for reference, please refer to the *Acc. Chem. Res.* **2021**, *54*, 1288), albeit rather underrepresented in this context. We further clarify this in the revised manuscript and we adjust the title to reflect more clearly that this is a perspective on the topic.

"This applies to different crystalline (*molecular*) *framework materials* defined by periodic assemblies of various nodes and (*molecular*) linkers that are relevant in this context (Fig. 1a),¹⁸ such as metal-organic frameworks (MOFs), covalent organic frameworks (COFs), and coordination polymers (CPs), but also hydrogen-bonded organic frameworks (HOFs), zeolites, periodic mesoporous organosilicas (PMOs), and hybrid perovskites, among others, that have been underrepresented as prospective DCMs." (page 2)

3. *The concepts of unidirectional and directional should be better distinguished and discussed, because they are recurrent in the manuscript: the theme is hot and deserves to be inscribed within neat boundaries to avoid confusing sentences. Experimental and theoretical results on this issue must be always distinguishable.*

Response: We are in agreement with the reviewer that this terminology can be misleading and we further clarify it in the manuscript. Specifically, *unidirectional motion* conventionally referred to *continuous directional motion* based on the previous reports. However, this might be misleading as it does not include all degrees of freedom, which we further clarify in the manuscript, complementing the previous examples.

"To this end, it is essential to *differentiate degrees of freedom* in such molecular frameworks (i.e., rotational, translational, and conformational, etc.), and distinguish between *random, restricted, controlled, and directional motion* (Fig. 1a)." (page 2)

"***continuous (oscillatory) directional (uni)directional motion*** is a prerequisite for a system to perform work comparable to macroscopic machines.³⁸ [...] Exemplary of such functionality is light-driven molecular motors, especially the overcrowded alkene-based molecular motors. Light-induced isomerization followed by a thermal relaxation step allows for a ratcheting mechanism that results in unidirectional 360° rotation.⁴¹ For instance, Danowski et al. embedded pyridine-functionalized motors into the backbone of MOFs while preserving unidirectional rotation.^{42,43} This provides the only example of unidirectional continuous responsive motion in crystalline solids. However, recent work by Gao et al. and Kathan et al. demonstrated how unidirectional rotation can result in the winding of polymer strands, resulting in macroscopic contraction of the molecular or polymer species.^{44,45}" (page 4)

42. Danowski, W. *et al.* Unidirectional rotary motion in a metal–organic framework. *Nat. Nanotechnol.* **2019**, *14*, 488.

Reviewer #2

This is an interesting perspective covering the field of functional dynamics in Framework materials. The subject coverage seems to be good and based on interesting from a wide range of authors. One concern is that the presentation is likely to miss the intended audience, which I assume should be relatively broad, covering chemists, physicist and materials scientists not in this field, as well as advanced undergraduate and graduate students. I will enumerate my comments and suggestions as follows.

Response: We are grateful for the positive assessment of the perspective. We share the concern about the reach of the audience, which motivates this perspective in the hope to engage a broader engagement across research fields, including chemists, physicists, and material scientists from both experimental and theoretical perspectives. We also appreciate constructive critical remarks that we address directly below.

1. Most figures are not as useful as they could be. The figure in the abstract does not convey a useful message. None of the figures illustrates what ADM framework materials might look like. Figure 1a is probably the most useful in the entire perspective, but it could be much better if it showed how those dynamic groups connect to the extended frameworks in Figure 1b.

Response: We appreciate the constructive remarks of the reviewer regarding the figures. Following the criticism, we revise the figures in order to better represent the concepts. This particularly refers to the former table of content figure that now better incorporates the classes of materials represented in Fig. 1. We also provide more information on how the dynamic groups connect to the extended frameworks in Fig. 1b.

Fig. 1. a) Structural representation of functional dynamics expressed through various degrees of freedom and the associated random, restricted (i.e., constrained, and controlled) directional motion in b) representative (molecular) framework materials and underlying dynamics of building blocks either inside the void space or incorporated inside the framework backbone.

2. How does an ADM zeolite look like? Are there any? It seems that what the authors may have in mind zeolites that undergo phase changes upon the absorption of some guests, like ZSM-5 and para-xylene?

Response: We discuss classes of framework materials that feature the potential to serve as dynamic crystalline materials, which should in this regard involve zeolites as well. We further refer to several representative examples, including light-switchable zeolite membranes where a photoswitching unit changes permeability or ion mobility in silicon-rich zeolite materials, as addressed in the revised manuscript.

"Similarly, cations in aluminium-rich zeolites exhibit dynamics within the cavities, blocking the transport of guest species.²⁴ In addition, light-controlled zeolite membranes have shown dynamic solid-state functionality. For instance, azobenzene incorporation into the zeolitic cavities was used to tailor its permeation,²⁵ illustrating the capacity of other various frameworks in realizing functional dynamics within pore cavities." (page 3)

24. Li, G. (Kevin) *et al.* Temperature-regulated guest admission and release in microporous materials. *Nat Commun* **8**, 15777 (2017).

25. Weh, K., Noack, M., Hoffmann, K., Schröder, K.-P. & Caro, J. Change of gas permeation by photoinduced switching of zeolite-azobenzene membranes of type MFI and FAU. *Micropor. Mesopor. Mat.* **54**, 15–26 (2002).

3. The use of the term “omnidirectional” seems inappropriate. Omnidirectionality implies that a given molecular entity can explore all the degrees of freedom that it has in the gas phase, or in a non-interacting liquid. The fact is that this is never the case when constrained environments reach the nanometer scale. Furthermore, the example shown with a floppy side chain linked to the framework cannot be omnidirectional as shown by the fact that it they are attached by a fixed permanently directional bond and cannot experience all rotational degrees of freedom. There must be a better term to describe this case. Constrained dynamics? Disordered dynamics?...

Response: We agree with the assessment of the reviewer that the term “omnidirectional” might be misleading, although we refer to it based on the previous literature reports. Considering the nature of the dynamics in the relevant examples, which does not reflect “omnidirectionality”, we refer to this as “restricted directional” instead of “restricted omnidirectional motion”. Thank you for your consideration.

” Among the most popular examples are molecular rotors predominantly based on para-substituted cyclic molecules (e.g., aromatics) with low torsional barriers featuring **restricted directional motion**.¹³” (page 3)

“While this illustrates the power of enhancing and facilitating specific dynamics in the solid state, the underlying motion is dictated by the molecular structure of the dynamic unit and its surroundings, driven by temperature and only to some degree directional in its nature.” (page 3)

4. Figure 2 is not useful if it is not more explicit as to what it all means, with the trajectories and dynamics of the corresponding elements. It is not clear what it means for a dynamic framework to be multidimensional. The message in Figure 3 is also poorly represented.

Response: Following the critical remarks, we have revised Figure 2 to better represent the underlying concepts. The main features of Figure 3 are now refined in Figure 2b. Thank you for your consideration.

Fig. 2. Schematic of the multidimensionality of the energy landscapes that can describe functional dynamics across (a) different length scales and systems, from the molecular level through the unit cell to the framework level, including host-guest systems, which (b) need to be complemented by temporal descriptors.

-
5. In page 3 second paragraph the authors refer to “strong” intermolecular pi-pi interactions as resulting in large barriers, which are not present in framework materials. In fact, pi-pi are not considered strong. Subsequently the authors offer a single example to make the comparison between molecular crystals and frameworks and they select a non-aromatic structure, bicyclo[2.2.2]octane. However, browsing through the literature one can find examples of molecular crystals with aromatic phenylenes rotators experiencing barriers lower than those of methyl groups in the gas phase, and one can also find open frameworks with rotators that have to surmount a very large intrinsic barriers, such as the iconic MOF 5. A recent JOC review article explicitly covers variations between different types of solids based on intrinsic barriers, free volume, and correlated dynamics that provides a general framework for analysis.

Response: We are in agreement with the reviewer that the reference to “strong intermolecular (e.g., π - π) interactions” has been misleading in this form, as referring to the interactions more comparatively, with π - π interactions provided only as an example which plays a role in mechanically interlocked molecules detailed further in the paragraph. We have thereby revised this statement to avoid confusion while also referring to a more general overview (*J. Org. Chem.* **2019**, *84*, 9835).

“While intermolecular (e.g., Van der Waals, π - π) interactions drastically enhance the barrier for rotation in densely packed molecular crystals, spatial separation of adjacent rotating molecules in molecular frameworks allows unrestricted rotational dynamics within a rigid 3D lattice.^{1–3,26}” (page 3)

26. Howe, M. E. & Garcia-Garibay, M. A. The Roles of Intrinsic Barriers and Crystal Fluidity in Determining the Dynamics of Crystalline Molecular Rotors and Molecular Machines. *J. Org. Chem.* **84**, 9835–9849 (2019).

6. In the last paragraph in Figure 4 the authors give a list of stimuli reported to influence the dynamics or properties of framework materials. It would be very useful for the interested readers to have references to those reports.

Response: Following the reviewer’s remark, we further outline the references to the paragraph (on page 5) detailing different stimuli used to control molecular dynamics in DCMs, which are further discussed below.

“While precise structural arrangements permit controlling molecular dynamics in DCMs, control over such motion is possible through *external stimulation* that enables functional response beyond temperature-controlled random motion and towards (uni)directional motion. To this end, different stimuli have been applied,^{35–37,49,50–51} including chemical processes (e.g., pH changes or chemical fuels),^{37,51} electrical charge,³⁷ light,^{35,43} and mechanical pressure.^{19,52}” (page 5)

7. The section on the assessment of functional materials across length scales in page 4 describes some of the analytical tools to measure dynamics in extended frameworks and other solids. It seems that this section is missing some interesting tools. This include the analysis of anisotropic displacement parameter (ADP) available from variable temperature single crystal x-ray diffraction, the information available from crystallographic disordered, which can be static or dynamic, and the use of frequency and temperature-dependent dielectric spectroscopy, and even inelastic neutron scattering for ultrafast dynamics. Giving the reader and idea of what method to use in what dynamic range would be greatly appreciated.

Response: We appreciate the reviewer’s input and we expand the discussion on the relevant analytical tools. In addition, we add a remark on the relevance of these methods within different dynamic ranges and complement with Figure 3 as an overview of some of the methods (inspired by previous reference reports).

“ This requires a *combination of theoretical and experimental techniques* that involve structural assessment as well as the analysis of optoelectronic characteristics of the materials via a range of spectroscopic and spectroelectrochemical techniques (Fig. 3).^{61–65} In particular, structural characteristics of DCMs are commonly analyzed using X-ray diffraction techniques in conjunction with theoretical models, as well as pair distribution function (PDF) analysis.^{35,48} Furthermore, the analysis of anisotropic displacement parameters (ADP) available from variable temperature single crystal X-ray diffraction is relevant, along with the use of frequency and temperature-dependent dielectric spectroscopy and inelastic neutron scattering for ultrafast dynamics.^{64,65} Modern time-resolved electron diffraction experiments allow following light-

induced processes on the nanoscale, drastically enhancing the time and space resolution of crystallographic methods.⁶⁶ Although these methods permit the assessment of crystallographic characteristics, establishing accurate modes in terms of extended structural complexity and dynamics levels is an ongoing challenge. This could be addressed by considering complementary techniques, such as Raman/IR and solid-state NMR spectroscopy, which remain underrepresented in this context despite the capacity to offer atomic-level insights.^{20–22} (page 7)

Fig. 3. Selected experimental (grey bars) and computational (white boxes) tools to study time-dependent processes in molecules and DCMs as a function of spatial and temporal resolution. This figure was inspired by ref. ^{63–65} and it does not provide a complete list of methods to study dynamics in framework materials.

8. The paragraph on dynamic cooperativity in dynamic crystals would not be complete without considering the 2020 Chemical Sciences assay on correlated motion and mechanical gearing by the Garcia-Garibay group.

Response: We are in agreement with the reviewer on the importance of the early work of Garcia-Garibay on molecular gearing and in particular the review summarizing some of the most representative examples of correlated motion in the solid state (*Chem. Sci.* **2020**, *11*, 12994). While we have already referred to this review article in our introduction (reference 3), we have now also pointed to molecular gearing and related examples in the paragraph discussing dynamic cooperativity in the solid state.

" Some of the earliest and most representative examples of solid-state molecular machinery with correlated motion are molecular gears transmitting motion from one element to another, including rotary arrays, molecular pumps, and motors.³" (page 5)

3. Liepuoniute, I., Jellen, M. J. & Garcia-Garibay, M. A. Correlated motion and mechanical gearing in amphidynamic crystalline molecular machines. *Chem. Sci.* **11**, 12994–13007 (2020).

9. To conclude, this reviewer feel that this perspective has great potential, but it needs a little work with the goal of becoming more scholarly, educational, and broad reaching.

Response: We thank the reviewer once more for their constructive input and hope that they find the revised manuscript more appropriate with respect to scholarly presentation and broader audience and impact.

Reviewer #3

*In the present perspectives article, some aspects of light-responsive framework materials are addressed. The focus is on one kind of molecular motors in COFs and MOFs. At first, it is not obvious why the focus was chosen, see below. The perspectives-article addresses a dynamic field. However, I cannot recognize a valuable contribution to the field. Recently, many excellent review articles were published, like Ref. 14, 15, <https://doi.org/10.1002/anie.201900666>, <https://doi.org/10.1021/acs.chemrev.1c00528> and <https://doi.org/10.1016/j.xcrp.2021.100544>. In comparison, the novelty, creativity and meaning of the present article are limited. I reckon that the perspective may eventually published, but I have my reservations that a journal like *Communications Chemistry* is appropriate. A more specific journal is more appropriate. Before resubmission, some issues need to be revised.*

Response: We thank the reviewer for their evaluation of the perspective article. We are in agreement with the reviewer that the research on light-responsive framework materials, particularly COFs and MOFs, is well documented in a number of recent review articles, which we also refer to in the manuscript. However, this perspective does not aim to provide a comprehensive review of the literature on this specific topic and goes beyond this research scope by addressing the question of functional solid-state dynamics across material classes, with a particular focus on framework materials. Apart from typical molecular frameworks, such as COFs and MOFs, this also involves coordination polymers, hydrogen-bonded organic frameworks, zeolites, periodic mesoporous organosilicas (PMOs), and hybrid perovskites that have been underrepresented in this context, as detailed in the manuscript. Moreover, we critically assess the criteria that define (amphi)dynamic solid-state materials and the methodology to assess them toward better fundamental understanding and practical applications. This unique perspective thereby distinguishes our work from the previous reports, which we believe makes it appropriate for *the Communications Chemistry* journal considering its broad research audience, and we appreciate the reviewer's consideration.

Following the critical remarks, we further clarify and emphasize the topical scope in the revised manuscript.

"In this perspective, we discuss a selection of recent developments of dynamic solid-state materials across material classes, outlining opportunities and fundamental and methodological challenges for their advancement toward innovative functionality and applications." (page 1)

"The research on DCMs develops at the intersection of several disciplines [...] which we do not review here in detail.^{3,8-16,17}" (page 1)

"In this perspective, we discuss *dynamic crystalline materials (DCMs)* from the standpoint of criteria that define them and opportunities presented to realize *functional dynamics* across classes of different *molecular framework materials*.¹⁸ In particular, we outline unique characteristics describing functional dynamics based on representative examples and consider methodological and other challenges for analyzing dynamic functions across length scales. Given the various existing terminologies, we describe these materials with responsive dynamic features and functionalities beyond lattice vibration and deformation as DCMs. We further address the importance of interdisciplinarity in providing a fundamental understanding of the pressing challenges, hoping to inspire researchers and new perspectives for this fascinating field." (page 1)

"This applies to different crystalline (*molecular*) *framework materials* defined by periodic assemblies of various nodes and (molecular) linkers that are relevant in this context (Fig. 1a),¹⁸ such as metal-organic frameworks (MOFs), covalent organic frameworks (COFs), and coordination polymers (CPs), but also hydrogen-bonded organic frameworks (HOFs), zeolites, periodic mesoporous organosilicas (PMOs), and hybrid perovskites, among others, that have been underrepresented as prospective ADMs." (page 2)

"While functional dynamics opens the way to unique phenomena, reaching beyond the conventional boundaries of research disciplines and material classes associated with solid-state dynamics is essential to overcome some pressing challenges.⁷⁸" (page 9)

We further address the critical points directly below.

-
1. *The article is filled with strong catchphrases without providing scientific insights. At least some solid scientific insights need to be provided.*

Response: We refer to research on various functional (amphi)dynamic crystalline materials and their characteristics throughout the manuscript while providing a broader perspective on this research topic and opening questions that are not commonly addressed in the related scientific communities. Following constructive inputs from the reviewers, we further clarified some of the terminologies and expanded on the discussion on some of the related material classes, methods, and examples, providing further insight. Please refer to the revised manuscript version with highlighted changes for your consideration. Thank you.

2. *The title does not match the content. A more accurate title is needed.*

Response: We believe that the title appropriately reflects the content of this perspective that primarily discusses functional dynamics in framework materials. To further clarify the scope of the content, following the reviewer's remark, we revise the title by specifying the reference to a perspective on the topic, by changing it to "A Perspective on Functional Dynamics in Framework Materials". We appreciate the reviewer's consideration and welcome further input.

"A Perspective on Functional Dynamics in Framework Materials" (page 1)

3. *The addressed dynamics focus on light-driven molecular motors of special kind. By checking the cited references, it is obvious the focus is on overcrowded-alkene-molecular-motors. This needs to be stressed in the text.*

Response: Following the reviewer's recommendation, we further specify in the manuscript that the most notable examples of the dynamic light-driven systems in the solids state refer to overcrowded alkene-based molecular motors. Apart from light-driven systems, we also provide examples on other stimuli-induced dynamic materials in the solid state.

"Exemplary of such functionality is light-driven molecular motors, especially overcrowded alkene-based molecular motors." (page 4)

"The resulting DCM response has predominantly focused on molecular building block dynamics, overcrowded alkene-based molecular motors in particular, without considering the impact on the extended material properties, especially in bulk materials." (page 5)

"To this end, different stimuli have been applied,^{35–37,49,50–51} including chemical processes (e.g., pH changes or chemical fuels),^{37,51} electrical charge,³⁷ light,^{35,43} and mechanical pressure.^{19,52}" (page 5)

4. *Figure 1 shows zeolites and perovskites. To term them framework materials is uncommon.*

Response: Although the term "framework material" might not be commonly used for zeolites and hybrid perovskites, it refers to various solid-state periodic assemblies comprised of nodes and (molecular) linkers (for reference on the terminology, please refer to *Acc. Chem. Res.* **2021**, *54*, 1288). This involves other classes of materials in addition to more commonly used molecular frameworks, e.g., COFs and MOFs, such as coordination polymers (CPs), hydrogen-bonded organic frameworks (HOFs), zeolites, periodic mesoporous organosilicas (PMOs), and hybrid perovskites. We further clarify this in the manuscript.

"This applies to different crystalline (*molecular*) framework materials defined by periodic assemblies of various nodes and (molecular) linkers that are relevant in this context (Fig. 1a),¹⁸ such as metal-organic frameworks (MOFs), covalent organic frameworks (COFs), and coordination polymers (CPs), but also hydrogen-bonded organic frameworks (HOFs), zeolites, periodic mesoporous organosilicas (PMOs), and hybrid perovskites, among others, that have been underrepresented as prospective DCMs." (page 2)

-
5. *The dynamics in zeolites and perovskite are not discussed in the text or, at least, mentioned. Especially the early work of light-switched zeolite membranes seems important to provide a time line for the field of dynamic solid materials.*

Response: In addition to the examples of hybrid perovskites that were previously provided in the perspective, following the reviewer's remark, we also refer to the examples of light-switched zeolite membranes. We thank the reviewer for the insightful remark.

"Such functional dynamics based on random molecular motion is relevant for other hybrid organic-inorganic framework materials, such as hybrid metal halide perovskites.^{18–22} They have demonstrated a unique capability to incorporate various organic species within their crystalline lattice that is otherwise primarily defined by the inorganic metal-halide framework.^{18–22} As a result, the overall structural properties and their functional dynamics could be controlled by an interplay of interactions between the organic and inorganic components in response to various external stimuli.²³ For instance, this enables reversible pressure-induced mechanochromism in these materials and, since recently, the integration of stimuli-responsive molecular components within the perovskite scaffold, opening a path towards multifunctional materials.^{17–18,23}" (page 3)

"Similarly, cations in aluminium-rich zeolites exhibit dynamics within the cavities, blocking the transport of guest species.²⁴ In addition, light-controlled zeolite membranes have shown dynamic solid-state functionality. For instance, azobenzene incorporation into the zeolitic cavities was used to tailor its permeation,²⁵ illustrating the capacity of other various frameworks in realizing functional dynamics within pore cavities." (page 3)

24. Li, G. (Kevin) *et al.* Temperature-regulated guest admission and release in microporous materials. *Nat Commun* **8**, 15777 (2017).

25. Weh, K., Noack, M., Hoffmann, K., Schröder, K.-P. & Caro, J. Change of gas permeation by photoinduced switching of zeolite-azobenzene membranes of type MFI and FAU. *Micropor. Mesopor. Mat.* **54**, 15–26 (2002).

6. *The authors should explain what they mean with the term "mechanism", used in Figure 3 and the abstract.*

Response: We have further revised Figure 3 to clarify the concepts discussed in the manuscript and highlight the importance of cooperativity and interdisciplinarity. In general, we refer to the term "mechanism" to describe the underlying dynamic processes by taking into consideration different types of motion, as discussed throughout the manuscript and selected examples, as illustrated below.

"However, the photoswitching mechanism strongly depends on the framework structure³⁴ and the switching of azobenzene moieties might exhibit a different mechanism when the switch is embedded within the framework through noncovalent complexation as compared to being attached to the framework backbone.³⁵" (page 4)

"Unlike the previous examples, these dynamic systems can transform specific energy inputs into unidirectional motion through a *ratcheting mechanism*.³⁸ Feng *et al.* showed that MOF-anchored rotaxanes³⁹ could facilitate the unidirectional transport of rings onto a polymer strand via redox cycling.⁴⁰" (page 4)

7. *The metaphors of the 4 climbers and the map in the abstract figure remain mysterious to me, even after spending some time trying to comprehend it. It should be revised.*

Response: Following the reviewer's remark, we further revise the former table of content figure for clarity, which is now included in the revised Figure 3 (now Figure 4) which highlights functional dynamics in various framework materials beyond lattice vibration and deformation, as well as the need for interdisciplinarity to address the challenges and enable opportunities for functional dynamics in framework materials across material classes and disciplines.

Fig. 4. Illustration of dynamics in framework materials (left) and the associated challenges and opportunities that require cooperative and interdisciplinary approaches (right).

8. The several statements in the conclusion are hard to follow: “One of the limiting factors refers to the toolbox for analyzing dynamics in the solid- state, especially from the perspective of in-situ structural characterization methods in response to external stimuli.” – The methods summarized in <https://doi.org/10.1021/acs.chemrev.1c00528> were used “While functional dynamics opens the way to unique phenomena, we believe that it is critically important to reach beyond conventional boundaries of research disciplines and material classes to overcome some of the pressing challenges.” – Which are?

Response: Following the reviewer’s remarks, we refer to the review summarizing time-resolved spectroscopy methods that are relevant in this context (*Chem. Rev.* **2022**, 122, 132–166). We further clarify what we meant by reaching “beyond the conventional boundaries of research disciplines and material classes”, which refers to those associated with solid-state dynamics. In particular, we imply considering other framework materials and relevant methodologies to analyze their characteristics.

“One of the limiting factors refers to the toolbox for analyzing dynamics in the solid state on the bulk, thin film, or single crystal level, especially from the perspective of in-situ structural characterization methods in response to external stimuli such as light.”⁷⁸ (page 9)

“While functional dynamics opens the way to unique phenomena, reaching beyond the conventional boundaries of research disciplines and material classes associated with solid-state dynamics is critically essential to overcome some pressing challenges.” (page 9)

78. Pattengale, B., Ostresh, S., Schmuttenmaer, C. A. & Neu, J. Interrogating Light-initiated Dynamics in Metal–Organic Frameworks with Time-resolved Spectroscopy. *Chem. Rev.* **122**, 132–166 (2022).

We thank you for your consideration.

REVIEWERS' COMMENTS:

Reviewer #1 (Remarks to the Author):

the manuscript is now suitable for publication. Especially, the title was changed and looks like more incisive.

Reviewer #2 (Remarks to the Author):

The revisions provided by the authors address all of the three reviewers comments well. The authors have taken ownership of their manuscript while addressing all constructive aspects of the three reviews. There is no doubt that a good manuscript has become a great manuscript and this reviewer is ready to offers his endorsements for publication of this work

Reviewer #3 (Remarks to the Author):

In the revised version, the authors responded to all questions from all referees. Many issues were solved, some issues remain:

1) The term “molecular frameworks” or “framework materials” do not include zeolites. It is arguable that it includes layered perovskites. The authors mention a review article (Acc. Chem. Res. 2021, 54, 1288) where hybrid perovskite is compared to MOFs. Indeed, this comparison is useful but the majority of the community considers layered or hybrid perovskites as inorganic–organic hybrid materials where many properties are dictated by inorganic components. There is no molecular framework permeating the perovskite. Zeolite materials are certainly not molecular framework materials.

Either the title, abstract and intro need to be revised or terms like “crystals” shall be used, that also match to zeolites or perovskite. As the major part deals with MOFs and COFs and dynamic processes in zeolites or perovskite are barely described it seems rational to remove zeolites and perovskite from Fig.1.

2) The length scale is addressed in Fig.3. MOFs with macroscopic length dynamics were recently discussed. See works of Masposh group and Woll group, <https://doi.org/10.1039/D2CC05813H> and <https://doi.org/10.1002/anie.202218052>. It seems that these framework dynamics are within the goals of “functional dynamics of molecular frameworks”. It should be discussed.

3) The difference of “functional dynamics of molecular frameworks” to “dynamic molecular crystals” should be highlighted. Maybe compare to <https://doi.org/10.1038/nchem.2547> and <https://doi.org/10.1021/ja411233p>.

I believe the manuscript will become suitable for publication after thoroughly solving these issues.

Reviewer Comments

Reviewer #1

The manuscript is now suitable for publication. Especially, the title was changed and looks like more incisive.

Response: We thank the reviewer for the support in improving the manuscript.

Reviewer #2

The revisions provided by the authors address all of the three reviewers comments well. The authors have taken ownership of their manuscript while addressing all constructive aspects of the three reviews. There is no doubt that a good manuscript has become a great manuscript and this reviewer is ready to offers his endorsements for publication of this work.

Response: We thank the reviewer for their positive feedback and support in improving the manuscript.

Reviewer #3

In the present perspectives article, some aspects of light-responsive framework materials are addressed. In the revised version, the authors responded to all questions from all referees. Many issues were solved, some issues remain:

Response: We thank the reviewer for their feedback and further address the critical points directly below.

- 1. The term “molecular frameworks” or “framework materials” do not include zeolites. It is arguable that it includes layered perovskites. The authors mention a review article (Acc. Chem. Res. 2021, 54, 1288) where hybrid perovskite is compared to MOFs. Indeed, this comparison is useful but the majority of the community considers layered or hybrid perovskites as inorganic–organic hybrid materials where many properties are dictated by inorganic components. There is no molecular framework permeating the perovskite. Zeolite materials are certainly not molecular framework materials. Either the title, abstract and intro need to be revised or terms like “crystals” shall be used, that also match to zeolites or perovskite. As the major part deals with MOFs and COFs and dynamic processes in zeolites or perovskite are barely described it seems rational to remove zeolites and perovskite from Fig. 1.*

Response: We thank the reviewer for the critical discussion on the nomenclature of the materials covered in our perspective. In our article, we aim to draw connections between various classes of materials, and there are numerous examples in literature that consider zeolites as framework materials, some of which are referred to in the article. In fact, zeolites feature all properties that are commonly related to framework materials (i.e., crystalline open 3D structures based on linkers and nodes as building blocks). We agree that “molecular framework material” might not provide an appropriate description in this case, and as such, we have put “molecular” in brackets and referred to a more inclusive term of “framework materials” throughout the text and figures. The intention of our article is to consider these materials from a different perspective of interest to a broader research community.

Furthermore, as the reviewer has pointed out, some properties of certain materials may be perceived differently when compared to other classes of materials, in particular when they are not commonly related (e.g., hybrid perovskites in comparison to MOFs). As such, “organic-inorganic hybrid materials” terminology could also be an accurate description for MOFs (although a recent IUPAC report has advised against such a definition; *Pure Appl. Chem.* **2013**, Vol. 85, pp. 1715–1724). One may argue that in some MOFs the function and structure primarily arise from the inorganic components, similar to hybrid perovskites, which does not exclude them from the “framework materials”. The perovskite structure type is a well-established framework motive and as such a great example of the framework materials discussed throughout this perspective. We highlight that in many of these materials local dynamic features are present regardless of the nature of the component they are built from, which should be considered from a more general rather than a materials-specific point of view. As such, we approached this perspective from an inclusive approach to illustrate the similarities with respect to dynamic features. We thus find that the depiction of the framework materials in Figure 1 is an accurate description of the topics covered throughout the manuscript. The fact that dynamic features of zeolites and hybrid perovskite are not extensively covered in this perspective reflects the relative number of studies on dynamics conducted with respect to these materials. We further hope that our perspective inspires researchers in the field of zeolites and perovskites to take inspiration from other framework materials to study and discover new dynamic aspects in these materials.

-
2. *The length scale is addressed in Fig.3. MOFs with macroscopic length dynamics were recently discussed. See works of Masposh group and Woll group, <https://doi.org/10.1039/D2CC05813H> and <https://doi.org/10.1002/anie.202218052>. It seems that these framework dynamics are within the goals of “functional dynamics of molecular frameworks”. It should be discussed.*

Response: We appreciate the reviewer’s reference to additional reports providing insights into framework dynamics. In our perspective, we aim to address local dynamics which are distinctly different from framework deformations or phase transitions that involve a dynamic change of the whole framework structure, such as those found in guest/temperature-responsive flexible frameworks. The recent review by Maspoch et al. given by the reviewer addresses macroscopic motion translated from crystal shrinkage or expansion. As such, we decided not to include this otherwise very interesting work as it does not address the aspects at the focus of our perspective. In contrast, the recent work by Henke et al. suggested by the reviewer addresses molecular switching via light stimulation that fosters macroscopic dynamics of a thin film. This is well within the scope of this paper, and we included this recent work as ref. 79. We thank the reviewer for bringing this to our attention.

“79. Jiang, Y. *et al.* Substrate-Bound Diarylethene-Based Anisotropic Metal–Organic Framework Films as Photoactuators with a Directed Response. *Angew. Chem. Int. Ed.* **62**, e202218052 (2023).”

3. *The difference of “functional dynamics of molecular frameworks” to “dynamic molecular crystals” should be highlighted. Maybe compare to <https://doi.org/10.1038/nchem.2547> and <https://doi.org/10.1021/ja411233p>.*

Response: To distinguish between functional dynamics of molecular frameworks and dynamic molecular crystals, in our manuscript we state that: “Achieving functional dynamics necessitates controlling the bottom-up assembly and the stimuli-responsiveness of the framework materials. The bottom-up (self-)assembly of intrinsically dynamic building blocks in molecular frameworks is a promising strategy toward this goal. It involves crystallizing dynamic building blocks into molecular crystals while maintaining intermolecular dynamics. Although many such dynamic molecular crystals are already known, their overall arrangement often lacks stability, and controlling their growth into extended functional materials remains challenging”. This implies that a critical property of framework materials is the ability to predicate their structure based on the selection of the underlying building blocks and templates (i.e., structure-directing agents). This is primarily based on the use of relatively strong intermolecular interactions (covalent, ionic, coordination bonds) that make up the framework backbone. Such concepts are very limited for the establishment of molecular crystals in which individual building blocks are relatively densely packed via rather weak van der Waals interactions. To further clarify this, we refer to several references on molecular dynamics in molecular crystals (see ref. 1,2,3,9) and we do not include more references. We have instead referred to appropriate ref. 1–2 in the section mentioned above and rephrased it as follows.

“It involves crystallizing dynamic molecular building blocks into molecular crystals while maintaining intermolecular dynamics.^{1,2} Although many such dynamic molecular crystals are already known, their overall arrangement often lacks stability, and synthetically controlling their structure, porosity, and growth into extended functional materials remains challenging.⁹ Materials such as MOFs, COFs, and hybrid perovskites have been considered as platforms in which dynamic building blocks can be co-assembled with secondary organic or inorganic components via coordination or covalent bonds to form robust 2D or 3D lattices. The framework structure can be predefined by the geometry of the building blocks, providing a vast yet defined structural space that can, to some degree, be predicted.”

We thank you for your consideration.